# Does the size of rewards influence performance in cognitively demanding tasks?

**Joachim A. Holst-Hansen, Carsten Bergenholtz** *

Department of Management, BSS, Aarhus University, Aarhus, Denmark

* cabe@mgmt.au.dk

## Abstract

Classic micro-economic and psychology theories propose different implications of monetary incentives on performance. Empirical studies in sports settings show that athletes generally perform worse when the stakes are higher, while a range of lab studies involving cognitively demanding tasks have led to diverging results, supporting positive, negative and null-effects of higher (vs. lower) stakes. In order to further investigate this issue, we present a pre-registered, randomized, controlled trial of 149 participants solving both anagrams and math addition tasks. We do not find a statistically significant effect of the size of the reward on neither performance, self-reported effort nor intrinsic motivation. We propose that future studies should contrast the potential impact of rewards on different kinds of task, e.g. compare tasks that solely require cognitive effort vs. tasks that require motor skills, as in sports.

## Introduction

How and if one should provide monetary incentives to individuals to improve their performance is a key question for the scientific disciplines economics and psychology, as well as managers in organizations. A general answer to this question is arguably unattainable, since different types of tasks, timeframes and organizational cultures seem to call for different reward systems [1–3]. When focusing on the nature of the tasks, meta-analyses reveal that in simple tasks, monetary incentives generally improve performance [3, 4]. This claim is in line with the micro-economic perspective, which argues that monetary incentives leads to increased motivation and effort, which in a task that depends on effort, will lead to higher performance [5, 6]. In contrast, a psychologist might argue that if the task is enjoyable, complex or embedded in an organization rather than in an artificial lab-experiment, the micro-economic explanation becomes inadequate, since intrinsic motivations will constitute a stronger influence on behavior, and these monetary incentives could crowd out intrinsic motivations [7].

Because most studies have merely compared offering some monetary reward vs. not offering a monetary reward at all, it is less clear if a higher monetary incentive has a different effect than a lower monetary incentive [5]. A manager can, in principle, modify the size of the incentive, which means the question is both of theoretical and practical interest. The proponents of the micro-economic perspective will argue that higher stakes lead to an even higher incentive to increase the effort, and if the task allows for further increase in effort, higher stakes should lead to a better performance than lower stakes [6]. Yet, too high stakes could also lead to a number of detrimental effects. For one, the opportunity to gain an exceptionally high reward

**Data Availability Statement:** All data files are available at the OSF repository: https://osf.io/fdgms/?view_only=3cc4847e870e417586af6d186881b8da database

**Funding:** The author(s) received no specific funding for this work.

**Competing interests:** The authors have declared that no competing interests exist.

could interfere with the participant's focus while performing the task [5], which could lead to a decrease of the participant's performance. Focusing on the magnitude of the potentially high reward might even make participants nervous, thus making them choke under pressure, further reducing the quality of their performance [5, 8]. Second, the relatively high stake might enhance the crowding out effect. If the task invites an intrinsic motivation, a higher reward can reduce the positive effect on effort from this intrinsic motivation. If this reduction is higher than the potential positive effect from a higher reward, the overall effect will be negative [7]. Third, if the given task can't be solved more efficiently than the level of performance a low monetary incentive allows, a higher stake can't have an additional positive effect due to a ceiling effect [3].

Data from sports events involving high stakes allows an examination of the impact of higher stakes on performance in real life scenarios. Studies ranging from basketball [9, 10], tennis [11], biathlon [12] to golf [13] find that higher stakes generally reduces performance of the participant. More specifically, in golf the effect leads to golfers putting worse when more money is at stake [13], while higher stakes implied diminished quality of basketball free throws in games compared to training [10] as well as worse free throws in crucial parts of the game [9]. Whereas [14] did not find a general effect in an analysis of penalty kicks in soccer, players did perform worse at home, an effect similar in kind to influence on the performance of biathletes [12]. Yet, two caveats should be added to these kinds of sports data. First, all these studies involve situations of fairly extreme pressure, exceeding the pressure an employee would usually meet in their daily work. Second, these studies do not rely on randomized trials, and hidden confounding factors might be shaping the results; e.g. the size of the audience, self-selection into these sports events or the higher status that beating other competitors implies.

Lab-studies allow for randomized controlled trials, where only the size of monetary rewards is varied. In the following we review studies relying on tasks that require cognitive skills and effort while solving some form of problem, thus excluding pure memory tasks [15] and tasks that do not require a cognitive but only a physical effort, such as typing [5]. Paradigmatic tasks relied upon are the Monty Hall problem, probabilistic challenges, anagram puzzles and additive math problems, i.e. cognitively demanding challenges most of which do not require particular experiences or educations.

Studies that have asked participants to solve various forms of cognitively demanding tasks have led to ambiguous findings; some studies find a positive effect on performance of higher stakes compared to lower [6, 16, 17], some find a negative effect [5, 18] while others find no clear effect when comparing high vs. low stakes [17, 19–22].

While the findings seem inconclusive, the reason might be systematic discrepancies in the used samples, type of tasks or size of rewards. Yet, these factors do not seem to create clear demarcation lines. A range of studies have relied on a task that required Bayesian reasoning, where some studies have found a positive effect of higher stakes [16, 17], whereas [20] found no effect of increasing the size of the rewards. Note, that while all three studies required Bayesian reasoning, one could argue that the Monty Hall problem used by Friedman [20] should be categorized as an insight problem, rather than requiring continuous probabilistic updating.

Furthermore, the exact same task (resembling an IQ test) has been used in Germany [18] and Israel [6]. The latter study offered 0, 0.1, 1, and 3 NIS (Shekel) in addition to a flatrate of 60 NIS, while the former study offered a similar reward structure (0, 1, 5 and 50 Eurocents) in addition to a flatrate of 5 Euros. Gneezy and Rustichini [6] found that the two highest reward levels performed statistically significantly better than the group not paid a piece-rate which in turn was better than the group paid 0.1 NIS as a piece-rate. Yet, in the replication study the very low piece-rate outperformed no piece-rate and high piece-rate, while other differences were not statistically significant [18]. Finally, the size of the reward is also not clearly associated

to higher performance. A high piece-rate reward of 50 cents vs. a low reward of 1 cent (factor of 50) did not improve performance in Miller & Estes' [22] study. In Ariely et al. [5] a factor of 10 generally led to a negative effect, whereas the same factor led to no significant impact on quality in Mason & Watts [19], while a factor of 3 and 1.5 has been shown not to lead to changes in performance [21]. Note that Mason & Watts [19] also examined if a higher reward improved productivity, i.e. how many tasks participants on the platform Amazon Mturk were willing to carry out. Participants thus had the opportunity to continue to solve Mason & Watts' [19] tasks or go to other, similar tasks, that were paid less/more. We only investigate performance in terms of relative number of correct solutions.

Given these inconclusive findings and the fact that relatively few studies showing no effect are available we worry that publication bias might be shaping the nature of the published results [23–25]. In order to contribute to the understanding of the implications of modifying monetary reward sizes, we present a pre-registered randomized controlled trial. We rely on a between-subject design, where primarily Danish participants where to solve some of the same cognitively demanding tasks (additive math and anagram) as in the most cited of all the above mentioned studies on higher vs. lower incentives [5]. Compared to former studies, we employed a medium level difference between higher vs. lower stakes. We expand on the experimental design in the next section.

The overall aim is to test if a higher piece-rate reward leads to a different performance level. More specifically, we test if substantially but not radically higher monetary rewards (factor of five, i.e. 2 DKK (0.26 Euros) vs 10 DKK (1.3 euros) for each correctly solved task) lead to a better performance when solving a cognitively demanding, yet relatively simple, task. Since the two overarching theoretical frameworks, micro-economics and self-determination theory as well as the empirical studies outlined above [5, 6, 16–22] offer ambiguous results, we will test the following competing hypotheses. The hypotheses as well as all statistical analysis are pre-registered, unless explicitly labelled exploratory (see https://osf.io/sb6ty/, an English version can be found in S1 Appendix).

*Hypothesis 1*: A larger piece-rate reward will lead to better performance.

*Hypothesis 2*: A larger piece-rate reward will lead to worse performance.

## Methods and data

We carry out a pre-registered, randomized controlled trial in order to identify if a higher or lower monetary reward influences the performance level. Participants solved two tasks in sequence, a math addition problem and an anagram. Both tasks are cognitively demanding and requiring substantial and continuous effort, without requiring an advanced education beyond primary school. Our study can be considered a conceptual replication of the aforementioned empirical studies, while being the most similar in kind to [5]; albeit differing in terms of the incentive size, being a between-subject rather than a within-subject study, relying on an increase in the number of tasks each participant engages with as well as the length of the task duration. The study was exempted from ethical approval by a Regional Committee on Health Research Ethics and was approved to be carried out at the university lab by the relevant Human Subjects Committee. Data is available at https://osf.io/fdgms/?view_only=3cc4847 e870e417586af6d186881b8da.

### Participants

Our sample consists of 149 participants, collected over two rounds with respectively 81 and 68 participants in each round. The latter round was added during the review process to solidify

the results and S4 Appendix contains descriptive statistics for each round. This sample size allows the identification of an effect size of r = 0.161 at a two-tailed p-value threshold of 0.05 if relying on a simple linear regression. We note that Funder and Ozer [26] categorize 0.1 as a small effect size and 0.2 as a medium effect size, while Gignac and Szodorai's meta-analysis [27] shows that 0.2 is a typical effect. Smaller effect sizes can be of interest if they accumulate [26], but in contrast to, e.g., a study on growth mindset [28] one needs to continuously re-invest the higher reward to generate the effect. The effect of rewards will thus not accumulate. We note that Ariely et al. [5] established an effect size of 0.351 for their equivalent math task, when comparing high vs. low rewards. This calculation is based on our re-analysis of their data (the authors kindly responded immediately to our request for their data), using a simple linear regression to assess how many more tasks were solved in the low reward condition.

The participants are primarily from Denmark (46.31% listed Danish as their mother tongue), 54.36% were female, and the average age was 25.05 (6.35 SD). These characteristics fit the general distribution of participants in the university's pool of lab participants. To our knowledge this is the first study of its kind to be carried out in Denmark. Pokorny [18] and Achtziger et al. [17] carried out their studies in Germany, which culturally speaking probably resembles Danish students the most, even though cultural differences between Germany and Denmark exist [29].

## Experimental design

**Exercises and survey.** We exposed participants to relatively simple, cognitively demanding tasks that mainly require effort, rather than a creative insight, without being pure memory tasks; math addition and anagram. We have chosen to employ two different tasks, which forces the participants to utilize different kind of skills, thus improving construct validity. The participants were given 10 minutes to solve two kinds of exercises (20 min in total), which each consisted of 50 tasks. All exercises and surveys were completed on paper. The first math addition exercise (cf. Fig 1) was to find the two numbers that constitute a sum of ten in a box with 12 numbers like below, an exercise identical to study two in [5]. Every box has exactly one solution.

In the next anagram exercise (cf. Fig 2) the participants were given 7–10 letters where the order had been randomized by an internet website [30], which uses the Fisher-Yates-Knuth shuffling algorithm to randomize the order of the letters. The participants had to construct a word in the English language using these letters. All the tasks in this exercise had at least one solution. Some tasks proved to have more than one solution, e.g. the letters "telsetr" can spell both letters and settler. No participants gave more than one answer to any tasks in this exercise.

Achtziger et al. [17] discovered that immediate feedback could play a moderating role, influencing the behavior after receiving the feedback. In our experiment, the participants received no external feedback during the experiment. However, most of the submitted answers

| 8.39 | 6.81 | 8.58 | | 8.39 | 6.81 | 8.58 |
|------|------|------|--|------|------|------|
| 3.65 | 9.68 | 3.33 | | 3.65 | **9.68** | 3.33 |
| 5.57 | 5.55 | 7.05 | | 5.57 | 5.55 | 7.05 |
| 0.32 | 4.54 | 1.09 | | **0.32** | 4.54 | 1.09 |

**Fig 1. Math addition task.** Example of a math addition task. The left side shows what participants saw, while the right side reveals the correct solution.

| Letters | Word |
|---------|------|
| osoemne |      |
| rrmbemee |     |
| telsetr |      |

**Fig 2. Anagram task.** Example of tasks. Note the third task has multiple solutions. Participants only had to find one and were not rewarded for finding more than one. No participants submitted more than one answer to any of the anagram tasks.

from the participants were correct and hence participants will probably have had some idea of their performance during the experiment.

The participants were presented the exercises one at a time, and in both kinds of exercises, the participants were required to solve the tasks in the order presented (the tasks were numbered 1–50). The participants were given ten minutes for each of the exercises and were not allowed to transfer time from one exercise to another. The tasks were solved on paper. The average number of tasks solved was 22.51 (9.42 SD), i.e. 22.51% of all tasks. No participant solved all tasks in any exercise, while 2 out of 149 participants solved more than 40 out of 50 math tasks. No one solved more than 40 (out of 50) anagram tasks. Therefore, performance in neither the low or high reward structure seems to face a ceiling effect cf. [3].

The allocated time span differs from the four minutes used by Ariely et al. [5]. We have chosen to engage the participants for a longer duration, since the longer time period can help identify smaller differences. For example, if participant A solves 0.1 task more than participant B per minute, a four-minute trial might end up with them having solved the same number of tasks while a ten-minute trial will reveal the difference. On the other hand, we wanted to avoid inducing cognitive fatigue by having the participants work for hours. The participants were told that a session (including additional surveys) would last forty minutes in total. No sessions exceeded this length, and no sessions were substantially shorter. Furthermore, since [5] relied on a within-subject design, participants ended up spending 16 minutes on the exercises within different incentives system, which is not very different from the 20 minutes our participants spend on solving the exercises.

The participants completed the addition exercise first. Hereafter they completed the TIPI personality survey [31] followed by the anagram exercise. Finally, they completed a survey aiming to measure three key variables identified in our literature review, i.e. effort (cf. micro-economics), intrinsic motivation (cf. self-determination theory) and focus [5]. The intention is to capture if these factors are correlated with the size of the reward. The questions are listed in S2 Appendix.

**Treatment.**   The treatment is the size of the reward, where the high stakes scenario involves a reward five times as big as the lower stakes scenario. In previous studies some have relied on much starker differences. Ariely et al. [5] employed a factor of 10 and 100 in some of their experiments, while both Miller & Estes [22] as well as Pokorno [18] relied on a factor of 50. In contrast, Achtziger et al. [17] doubled the reward in the high stakes scenario. Overall, compared to former studies a factor of five is at a medium level. We have tried to balance the dual aims of external validity and avoid the risk that an affect disappears in statistical 'noise'. A factor of five is somewhat closer to a realistic managerial scenario than if the high reward is 50 or 100 times higher than the low reward. Even if a factor of 50 influenced behavior, it would be unlikely that such an intervention would be economically beneficial to implement.

The participants in the condition with a low reward got 2 DKK (0.26 Euros) per correctly solved task, while participants in the high reward scenario received 10 DKK. 30 DKK were

added to the money earned from solving the tasks in order to ensure that participants reached the minimum baseline pay of 40 DKK that participants in the lab have to receive to participate in a study. Only 3 participants solved too few tasks and had to be boosted to 40 DKK. The reward was taxable income and reported to tax authorities by the university. The best paid participant received 470 DKK (63 Euros) for forty minutes of effort, which equals an hourly wage of 705 DKK, before taxes. This is a high wage, but it is not unlikely that some of the participants will earn a similar wage at some point in their life, especially considering most of them are university students.

The following paragraph is the information regarding reward size given to the participants with the low [high] reward:

> *Your expected earnings for participating in the study are between 40 [40] and 230 [1.030] Danish crowns. You will be rewarded with a fixed amount of 30 [30] DKK. For each correctly solved task, 2 [10] DKK will be added to your payment. You cannot earn less than 40 [40] DKK. The reward for solving one task is the same in both types of exercises.*

The participants were not informed that the experiment was about pay and performance, and thus were not aware that other participants were exposed to a different reward structure. Participants were given the following piece of information regarding the aim of the experiment when signing up: "The aim of the study is to gain more knowledge as to how performance can be influenced by contextual factors outside the participant's control."

Deciding not to inform participants about the variation in reward structure also implies that we relied on a between-subject design instead of the within-subject design in Ariely et al. [5]. Earlier research showed that changing reward structure could influence behavior in the second round of exercises [16]. Furthermore, in a within-subject design one risks that participants learn more efficient problem-solving strategies and become better over time. A between-subject design thus reduces the risk of 'noise' from learning and switching effects.

## Randomization

Two sessions were run on any given day, one starting at 09:00 and one starting at 10:00. The same time of day was chosen since cognitive fatigue can influence students' performance on standardized tests [32], and therefore running experiments at many different times of the day would induce unnecessary variation. A random number generated by Excel was used to decide if the first session would be run with the low or the high reward. The other session was then run with the other reward structure. No sessions were run Saturday or Sunday, and two days involved only one session, due to lack of participants. The slots were posted on the laboratory's website for participants and the participants could then choose to participate in the experiment.

## Analysis

The analysis has been preregistered (https://osf.io/sb6ty/) following the template from AsPredicted [33]. An English version can be found in S1 Appendix.

### Variables in primary analysis

Non self-explanatory variables are described in the following.

**Performance (dependent variable).**   Performance is the sum of the number of solved tasks in the adding task and the anagram task. The sum is chosen as the dependent variable instead of the two exercises being analyzed separately. This pre-registered approach is selected

in order to reduce the degrees of freedom and because the exercises are chosen to test the same thing; influence from size of reward on performance when solving cognitively demanding tasks.

**Conscientiousness (control variable).**   Conscientiousness has consistently been shown to be correlated with job performance [34], which is why it is included as a control variable, and captured via the TIPI [31]. We selected this short personality inventory due a worry about the potential cognitive fatigue that the long tests might imply and the fact that it is a control variable, not a main predictor. See S3 Appendix for an overview of the questions.

### Variables in secondary analysis

In order to further investigate how the reward size might influence behavior, we also assess if self-reported measures of effort [6], intrinsic motivation [7] and focus [5] vary depending on the reward structure the participants have been exposed to. These three variables are described in the following.

**Intrinsic motivation and effort.**   In order to measure intrinsic motivation (cf. self-determination theory) and effort (cf. microeconomics) we have used questions based on IMI [35], but worded according to our setting. For reference, [36] report a Cronbach's alpha of 0.78 for the interest-enjoyment scale (our measure of intrinsic motivation) and 0.84 for effort. Four questions capture the self-perceived effort that the participant put into solving the exercises, while four questions capture the self-perceived interest and enjoyment (i.e. intrinsic motivation). All questions are listed in S2 Appendix. In order to calculate an intrinsic motivation score, answers to question 2, 4 (reversed), 7 (reversed), and 8 are averaged (cf. the appendix). In order to calculate an effort score, answers to question 1, 3 (reversed), 5 (reversed), and 6 are averaged (cf. the appendix).

Since three hypotheses are tested in the secondary analysis, the alpha level is adjusted using Bonferroni's method. This means that the alpha level is 0.0167. We have tested for difference in mean using Welch's t-test instead of Student's t-test. This test's risk of type 1 error is closer to the significance level when the variances are unequal compared to both Student's t-test and a choice between Student's t-test and Welch's t-test based on a preliminary test for equality of variances [37].

**Focus.**   In order to measure how focused the participants were, three questions have been developed. These are the three last questions from the questionnaire shown in S2 Appendix. These questions aim to capture if the reward distracted the participants in their task solving. The average from these three answers (with the last question reversed) is the focus score for a participant.

## Results

### Descriptive statistics

Fig 3 illustrates how there were no substantial differences between the individuals in the two conditions. Out of the 149 participants two persons did not inform their age, and two persons did not fill out the survey designed to measure effort, intrinsic motivation, and focus.

### Primary analysis

The data has been analyzed using a multiple linear regression, using the size of the reward (low or high) as an independent variable, while gender and conscientiousness have been added as covariates. The only pre-registered dependent variable is the total number of solved tasks, which allows us to investigate if a high or low reward influences performance.

| Variable | Sample | |
|---|---|---|
| Female | Entire sample | 54.36% |
| | High reward | 54.55% |
| | Low reward | 54.17% |
| Danish | Entire sample | 46.31% |
| | High reward | 38.96% |
| | Low reward | 54.17% |
| Age | Entire sample | 25.05 (6.35) |
| | High reward | 25.07 (6.97) |
| | Low reward | 25.04 (5.69) |
| Performance | Entire sample | 22.51 (9.42) |
| | High reward | 23.65 (8.45) |
| | Low reward | 21.29 (10.28) |
| Conscientiousness | Entire sample | 5.01 (1.27) |
| | High reward | 5.02 (1.21) |
| | Low reward | 4.99 (1.33) |
| Effort | Entire sample | 5.97 (0.98) |
| | High reward | 5.84 (1.03) |
| | Low reward | 6.11 (0.91) |
| Intrinsic motivation | Entire sample | 5.11 (1.18) |
| | High reward | 5.10 (1.11) |
| | Low reward | 5.11 (1.25) |
| Focus | Entire sample | 3.49 (1.30) |
| | High reward | 3.61 (1.25) |
| | Low reward | 3.38 (1.35) |

**Fig 3. Descriptive statistics.** Numbers in parentheses are standard deviations.

Homoskedasticity has not been assumed, and thus robust standard errors have been used. The regression was estimated using OLS with the following regression equation:

$$total\ number\ of\ solved\ tasks = \beta_0 + \beta_1 * large\ reward + \beta_2 * female + \beta_3 * conscientiousness + u$$

Fig 4 reveals a p-value of 0.12, and thus no strong support for either a positive or negative effect from a larger reward. However, we note that the confidence interval is rather wide, and a positive effect is still contained in the confidence interval.

We provide an additional, exploratory analysis, since some participants on the first day of sessions (two sessions, 16 participants) did not solve the tasks in the order presented, despite this being noted as an explicit requirement in the written instructions. In the following analysis

| | | | | | |
|---|---|---|---|---|---|
| Number of obs | | | | | 149 |
| F(3, 145) | | | | | 2.63 |
| Prob > F | | | | | 0.05 |
| R-squared | | | | | 0.05 |

| Performance | Coef. | Robust Std. Err. | t | P>\|t\| | [95% Conf. Interval] |
|---|---|---|---|---|---|
| Highreward | 2.39 | 1.53 | 1.57 | 0.12 | -0.63 ; 5.42 |
| Female | -0.44 | 1.51 | -0.29 | 0.77 | -3.41 ; 2.54 |
| Conscientiousness | -1.30 | 0.61 | -2.13 | 0.04 | -2.52 ; -0.09 |
| Constant | 28.04 | 3.68 | 7.61 | 0.00 | 20.76 ; 35.32 |

**Fig 4. Main regression.** Result from main regression.

answers from these two sessions were corrected as if the instruction had not been given. In the remaining sessions this instruction was emphasized, and these answers were corrected with this instruction in mind. We include a dummy for the first day of sessions, in order to control for the difference, which leads to the following equation:

$$total \ number \ of \ solved \ tasks = \beta_0 + \beta_1 * large \ reward + \beta_2 * female + \beta_3 * conscientiousness + \beta_4 * Monday + u$$

The results in Fig 5 are almost identical to the former model, showing a slightly higher p-value (0.16) and very similar confidence interval. Thus, including the dummy does not change the conclusion, and we again cannot reject the null hypothesis that size of reward does not influence performance.

## Secondary analysis

All three theories identified in the introduction rely on a mediating, explanatory variable. We therefore also analyse if the size of reward influenced effort (mediator in microeconomics), intrinsic motivation (mediator in self-determination theory), or focus (mentioned as a potential meditator by [5]). This allows us to test if we might be unable to measure the effect of reward size on performance, but can identify a change in the level of effort, motivation or focus.

| | | | | | |
|---|---|---|---|---|---|
| Number of obs | | | | | 149 |
| F(4, 144) | | | | | 2.51 |
| Prob > F | | | | | 0.04 |
| R-squared | | | | | 0.05 |

| Performance | Coef. | Robust Std. Err. | t | P>\|t\| | [95% Conf. Interval] |
|---|---|---|---|---|---|
| Highreward | 2.19 | 1.55 | 1.41 | 0.16 | -0.88 ; 5.26 |
| Female | -0.53 | 1.51 | -0.35 | 0.73 | -3.52 ; 2.46 |
| Conscientiousness | -1.24 | 0.63 | -1.98 | 0.05 | -2.48 ; 0.00 |
| Monday | 2.00 | 2.26 | 0.88 | 0.38 | -2.47 ; 6.47 |
| Constant | 27.65 | 3.76 | 7.35 | 0.00 | 20.22 ; 35.09 |

**Fig 5. Regression with added covariate.** Result from regression with added covariate.

|  | *Focus* | *Intrinsic motivation* | *Effort* |
|---|---|---|---|
| t Stat | -1.07 | 0.05 | 1.70 |
| P(T<=t) two-tail | 0.29 | 0.96 | 0.09 |
| 98.33 % confidence interval (low-high) | -0.75 ; 0.29 | -0.46 ; 0.48 | -0.12 ; 0.66 |
| Cronbach's α | 0.55 | 0.76 | 0.74 |

**Fig 6. Focus, motivation and effort related to reward size.** t-tests for equality of means, comparing high vs low rewarded sizes.

The descriptive data only contain 147 observations since two participants chose to not fill out the questionnaire aimed at measuring these things.

The data do not show a significant effect from size of reward on either focus, intrinsic motivation, or effort at the 0.0167 threshold. Effort has the lowest p-value (p = 0.09), however the participants who received the low reward exerted *more* effort according to data–the opposite of the prediction offered by classical microeconomics.

The data can be exploratory re-analysed excluding the results from the first two sessions. However, this does not result in a statistically significant difference in any of the tests (see S4 Appendix). We also note that the low Cronbach's α for the questions measuring focus (0.55, cf. Fig 6) imply the associated t-test should be interpreted with caution.

## Exploratory analysis

We tested whether the self-reported effort, intrinsic motivation, and focus influenced performance, irrespective of the reward size.

Fig 7 shows that both effort (p < 0.01 for both models) and intrinsic motivation (p = 0.018 and 0.020 for model 3 and 4 respectively) are positively and significantly related to performance, as predicted by classical microeconomics and self-determination theory. Participants that performed well thus self-reported higher levels of effort and intrinsic motivation than those that did not do well, corroborating that these factors are indeed important for the tasks.

|  | Effort | | Intrinsic motivation | | Focus | |
|---|---|---|---|---|---|---|
|  | 1 | 2 | 3 | 4 | 5 | 6 |
| **Mediator** | 2.31*** | 2.36*** | 1.70** | 1.68** | 0.62 | 0.64 |
|  | (0.77) | (0.76) | (0.71) | (0.71) | (0.61) | (0.61) |
|  |  |  |  |  |  |  |
| **Monday included as covariate** | NO | YES | NO | YES | NO | YES |
| **Gender and conscientiousness included** | NO | NO | NO | NO | NO | NO |
| **Constant included** | YES | YES | YES | YES | YES | YES |
| **Number of obs** | 147 | 147 | 147 | 147 | 147 | 147 |
| **R-squared** | 0.06 | 0.07 | 0.04 | 0.05 | <0.01 | 0.02 |

**Fig 7. Effort, motivation and focus related to performance.** Relationship between performance and mediators without extra covariates. *, **, and *** mean the variable is statistically significant at the 10%, 5%, or 1% level.

| | Effort | | Intrinsic motivation | | Focus | |
|---|---|---|---|---|---|---|
| | 1 | 2 | 3 | 4 | 5 | 6 |
| **Mediator** | 2.65*** | 2.67*** | 1.62** | 1.61** | 0.58 | 0.60 |
| | (0.79) | (0.78) | (0.69) | (0.69) | (0.62) | (0.62) |
| | | | | | | |
| **Monday included as covariate** | NO | YES | NO | YES | NO | YES |
| **Gender and conscientiousness included** | YES | YES | YES | YES | YES | YES |
| **Constant included** | YES | YES | YES | YES | YES | YES |
| **Number of obs** | 147 | 147 | 147 | 147 | 147 | 147 |
| **R-squared** | 0.11 | 0.12 | 0.08 | 0.08 | 0.04 | 0.05 |

**Fig 8. Effort, motivation and focus related to performance, including covariates.** Relationship between performance and mediators with extra covariates. *, **, and *** mean the variable is statistically significant at the 10%, 5%, or 1% level.

A statistically significant effect of focus on performance could not be detected (p = 0.318 and p = 0.294). However, this might be due to the fact that questions aimed at measuring focus did not fully capture this construct, cf. the relatively low Cronbach's α. These results are robust to model specifications (see Fig 8 which includes covariates) and the exclusion of data from the first two sessions (see S4 Appendix).

## Discussion

Results from nine former studies on the comparative impact of high vs. low rewards are remarkably mixed, since one can find support for all kinds of effects; positive, negative and no effect. We worry that publication bias might skew the available results, since documenting a null effect is difficult and have a higher risk of not being published [23, 38], while documenting small effects require substantial sample sizes.

In our pre-registered randomized, controlled trial we do not find strong evidence that supports higher stakes reduce or improve performance. The absence of evidence is not evidence of an absence of an effect from a higher reward, of course, but given our sample size we could have identified an effect size of r = 0.161 if relying on a simple linear regression (cf. [26]). The correlation between high reward and performance was 0.125 in our study, and with this correlation we would have required a sample size of 247 to obtain a p-value below 0.05, if relying on a simple linear regression. While we cannot rule out the existence of a small positive effect of high rewards, our study reduces the prior belief one should have in the replicability of former, relatively large effect sizes, in either direction. Furthermore, a small non-cumulative effect would generally not be economically efficient to implement for a manager. To illustrate, even if we assume that the difference our data implies are not just random fluctuations, the high rewards merely lead to approximately two extra solved tasks, out of an average of about 22 tasks solved. Yet, a manager would have to pay a factor of 5 to gain such a relatively small improvement. We should add that higher stakes would be economically beneficial for the one receiving the higher reward.

Additional analysis supports the interpretation that we do not find strong support for reward structure influencing the participants in our sample. Micro-economics argues that a

higher reward should lead to a higher performance since the reward structure should influence the effort, while self-determination theory argues that a higher reward could harm intrinsic motivation. However, the size of the reward did not have a statistically significant effect on neither the self-reported level of effort nor intrinsic motivation, while both effort and intrinsic motivation was significantly positively related to performance. Thus, we do not find strong support for the size of the reward shaping the dependent variable, or the mediating variables they are supposed to influence.

Overall, while former studies have shown that a piece-rate payment system generally leads to higher performance compared to flat-rate payments when individuals are solving relatively simple tasks [3], we find no clear behavioral effect from manipulating the reward size. Since our data did not imply a risk of a ceiling effect, one could speculate that getting some kind of monetary reward is a sufficient motivator for participants in a lab-study, while having the opportunity to gain relatively big rewards do not lead participants to choke due to pressure [5].

Yet, a range of studies from the world of sports repeatedly indicate that large reward differences, tend to lead to high stakes reducing performance [9–13]. This discrepancy between sports and cognitive tasks in the lab might simply be the difference between a randomized, controlled trial and observational studies, which cannot exclude potentially confounding factors, such as the public scrutiny and the status competition that sports events also imply. Ideally one would have data on sports events with only a single competitor or without an audience. However, in addition to these confounding factors, we also find good reasons to believe that different theoretical mechanisms are in play in the various settings. All sports disciplines studied require not only a basic cognitive effort (as in the typical lab-task) but also a concentrated, physical motor effort; e.g. when completing a golf putt or a basketball free throw [9, 10, 13]. These various skills arguably have a different evolutionary history: "We are all prodigious Olympians in perceptual and motor areas, so good that we make the difficult look easy. Abstract thought, though, is a new trick, perhaps less than 100 thousand years old. We have not yet mastered it." [39 p. 2]. One can therefore argue that we should expect performance differences depending on the skills required in the given setting. For example, being able to drive a golf ball relies on motor skills, while a golf putt requires not only motor skills but also a cognitive effort and ability to assess the length of the putt and slope of the green.

We consider it a promising avenue for future research to disentangle these variables. We propose that future studies should investigate the contrast between a) simple, cognitively mundane tasks [5], b) insight tasks [40, 41], c) tasks only requiring motor skills (e.g. a golf drive) and d) tasks that require motor skills as well as cognition (e.g. a golf putt). If one relies on a typical pool of lab-participants, one also avoids the potential selection effect of the very experienced sports players that participate in high performance sports. We expect that tasks solely requiring high motor skills should see less of a negative effect of higher stakes compared to tasks that require motor skills as well as cognition. High stakes in insight tasks in particular and cognitively very demanding tasks in general could lead to worse performance levels compared to lower stakes scenarios. This negative effect is argued to be due to reduction in intrinsic motivation, ability to focus and that the reward structure can lead to a more conservative search strategy [1] inhibiting the opportunity to find good solutions to the task at hand. Finally, all these investigations could involve a competitive element, as in sports, or not, in order to further identify the extent of this effect.

To sum up, we did not identify an effect of the size of the stake involved, when solving a mundane, cognitive task. We worry that the lack of articles not showing an effect might reflect a file-drawer bias [23, 25]. Only by providing public access to studies that cannot document an effect, can we update the relevant prior one should have concerning the potential impact of large reward sizes [42].

## Supporting information

**S1 Appendix.**
(DOCX)

**S2 Appendix.**
(DOCX)

**S3 Appendix.**
(DOCX)

**S4 Appendix.**
(DOCX)

## Author Contributions

**Conceptualization:** Joachim A. Holst-Hansen.

**Data curation:** Joachim A. Holst-Hansen.

**Investigation:** Joachim A. Holst-Hansen.

**Methodology:** Joachim A. Holst-Hansen, Carsten Bergenholtz.

**Project administration:** Joachim A. Holst-Hansen, Carsten Bergenholtz.

**Supervision:** Carsten Bergenholtz.

**Writing – original draft:** Joachim A. Holst-Hansen.

**Writing – review & editing:** Joachim A. Holst-Hansen, Carsten Bergenholtz.

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
