## [Decision Letter · Decision Letter 0]

14 Feb 2020

PONE-D-19-35879

Does the size of rewards influence performance in cognitively demanding tasks?

PLOS ONE

Dear Mr. Bergenholtz,

Thank you for submitting your manuscript to PLOS ONE. After careful consideration, we feel that it has merit but does not fully meet PLOS ONE’s publication criteria as it currently stands. Therefore, we invite you to submit a revised version of the manuscript that addresses the points raised during the review process.

As you can see, the two reviewers make different points in some respect, but I see two things that I consider of high importance. First, it´s the sample size. Both reviewers indicate either directly or rather indirectly, that there actually might be an effect once the sample size is big enough. I agree with this request - if the sample size is just too small to find an effect, your null-results would be an artifact. So I strongly recommend to increase the sample size - at least doubling it, but more would be better.

Secondly, Reviewer 2 made quite some comments about your argumentation with respect to what this contributes to the literature and why. Please go over these comments (and all others!) carefully and see whether and how you can address them.

Overall, I agree with Reviewer 1 that the paper is in general well written and well organized. So please take care when answering the requests from Reviewer 2 to keep that nice general outline!

We would appreciate receiving your revised manuscript by Mar 30 2020 11:59PM. To enhance the reproducibility of your results, we recommend that if applicable you deposit your laboratory protocols in protocols.io, where a protocol can be assigned its own identifier (DOI) such that it can be cited independently in the future. For instructions see: http://journals.plos.org/plosone/s/submission-guidelines#loc-laboratory-protocols

We look forward to receiving your revised manuscript.

Kind regards,

Christiane Schwieren, Dr.

Academic Editor

PLOS ONE

Journal Requirements:

2. Please provide additional information about the participant recruitment method and the demographic details of your participants, such as, when relevant, the recruitment date range (month and year), a description of any inclusion/exclusion criteria that were applied to participant recruitment, a table of relevant demographic details, a statement as to whether your sample can be considered representative of a larger population, a description of how participants were recruited, and descriptions of where participants were recruited and where the research took place.

3. Your ethics statement must appear in the Methods section of your manuscript. If your ethics statement is written in any section besides the Methods, please move it to the Methods section and delete it from any other section. Please also ensure that your ethics statement is included in your manuscript, as the ethics section of your online submission will not be published alongside your manuscript.

5. Please ensure that you refer to Figure 1 and 2 in your text as, if accepted, production will need this reference to link the reader to the figure.

Reviewers' comments:

Reviewer's Responses to Questions

**Comments to the Author**

1. Is the manuscript technically sound, and do the data support the conclusions?

Reviewer #1: Yes

Reviewer #2: Partly

2. Has the statistical analysis been performed appropriately and rigorously? 

Reviewer #1: Yes

Reviewer #2: Yes

3. Have the authors made all data underlying the findings in their manuscript fully available?

Reviewer #1: Yes

Reviewer #2: Yes

4. Is the manuscript presented in an intelligible fashion and written in standard English?

Reviewer #1: Yes

Reviewer #2: Yes

5. Review Comments to the Author

Reviewer #1: Congratulations on a paper that you wrote in a remarkably crispy, clear and concise way. Everything that should be there is there - and nothing more (which is also an art). I likey all about the paper with one exception - the sample size!

Why would you rund such a nice experiment with just 81 subjects - funding cannot really be an issue; neither can missing subjects be - on any university campus shoudl you be able to get hundreds of subjects. Why is that important - because a comparison of 40 vs 40 subjects probably MISSES effects that are there - your results deliver a p-value of .33; but properly powered (400 subjects total) you would be able to deliver a much stronger message (be it a significant difference or not).

Hence, I suggested accepting the manuscript as it is; but if the other referee(s) or the edior "force" you to increase sample size - do that, it may be worth it.

Reviewer #2: This paper considers the issue of whether higher stakes lead to better performance. It is a somewhat controversial topic. But to me the main issue is at the end of the range, with very high or very low incentives. This study has payoffs in one case multiplied by five and I am not sure this captures the issue. But it appears the experiments were conducted properly, so it would appear to be rigorous. I just don’t see what we learn from this study.

Abstract, 2nd sentence: Seriously? Then why do teams pay pro athletes so much? Can’t really be true as stated.

Introduction, 2nd paragraph: I really don’t agree with this. Higher pay clearly typically leads to higher effort. The issue of interest is really just at the extremes of the pay range.

p. 9, middle: These are examples of extreme pressure, not just pressure. And the golf study is just one study. One wonders how many other studies found no effect.

p. 9, lower: It's really quite silly to say that payment per se leads to worse performance.

p. 9, penultimate full paragraph: I'm not sure this is an entirely fair review of the literature.

First line of paragraph before Hypothesis 1: But this may well depend on where in the range you are.

Hypotheses 1 and 2: Over the entire range?

p. 12, lower: The between- versus within-subject could easily have mattered. Good that you mention this later.

p. 13, upper: I don't know that a factor of five will matter in this range for this type of task.

p. 14, Performance: This is still dubious. I'd also like to see separate analyses.

p. 14, Intrinsic motivation: I’m not really keen on any of these psych measures, but presumably this was stated in the pre-analysis plan.

Before Figure 4: But it does in fact look like there is a positive difference. I wouldn’t necessarily be claiming a big victory here.

Secondary analysis: 1) I’m not sure how much I trust these self-determination measures. 2) With the Cronbach number so low, I would just ignore this.

End of first paragraph of Discussion: I think the results are unclear here. Yes, they are cautionary.

First line of penultimate paragraph: I agree with this. It's interesting.

6. PLOS authors have the option to publish the peer review history of their article (what does this mean?). If published, this will include your full peer review and any attached files.

Reviewer #1: Yes: Juergen Huber

Reviewer #2: No

---

## [Author Response · Author response to Decision Letter 0]

19 Aug 2020

Dear editor

We appreciate the constructive reviewer comments as well as your clear and explicit guidance for how to integrate the provided suggestions. Also, we would like to thank PLOS ONE for kindly granting us additional time, in order for us to collect more data. In the following we first address your editorial comments, where after we address each reviewers’ comments one by one.

We agree that increasing the sample size greatly benefits our ability to provide stronger evidence for the inferences we intend to draw. Following your recommendation, our plan was to double the sample size. Due to Covid-19 our lab was closed for about 3 months in the Spring of 2020. After reopening, we managed to increase our sample size from 81 to 149, before our university had to close down the lab again, forcing us to cancel the last session we had planned. However, the results remain insignificant. Furthermore, given the current correlation we would need to further increase the sample size substantially, in order to document a significant positive effect of higher stakes; to about 247, as explained in the paper. In other words, the increased total sample size we now rely upon allows us to say that the sample should have shown a correlation of r = 0.16 in order for the results to be statistically significant with a simple linear regression. In other words, we find no significant result, and if there is an effect, it is relatively small – and certainly economically inefficient.

We’d also like to emphasize that our sample size is at the high end, compared to the 9 studies we refer to in our theoretical framework. As in other related theoretical fields, effect sizes of the impact of high (vs. low rewards) seem to turn out smaller than original studies indicate. We should also add that the performance in each condition in the newly collected data is almost identical to our former results, indicating that we should not worry that behavior was different in the Summer of 2020, compared to pre-pandemic times (see S4 Appendix for the relevant table). Our country was functioning quite normally in the relevant period, btw. 

All findings, tables and figures have been updated, integrating the new data and points we make in the above section. In accordance with your suggestion, we have maintained the general outline of the paper, but also adjusted some phrasings to ensure that we write, for example; “we do not find a significant difference between the two conditions”, rather than, for example: “our results show that a higher stake does not lead to increased performance”. In essence, we report a null-finding, yet we also document that given our setup, one requires fairly large sample sizes, if one wants to showcase robust significant results.

Response to Reviewer 1

Reviewer comment: Congratulations on a paper that you wrote in a remarkably crispy, clear and concise way. Everything that should be there is there - and nothing more (which is also an art).

Reply: Thank you very much for your gracious remarks on our paper!

Reviewer comment: I like all about the paper with one exception - the sample size!

Reply: We agree on your concern about the sample size and appreciate the opportunity to enhance the strength of our inferences. We therefore planned to double the sample (following the recommendation by the editor). However, due to Covid-19 our lab was closed for about 3 months in the Spring of 2020. After reopening, we managed to increase our sample size from 81 to 149, before our university had to close down the lab again, forcing us to cancel the last session we had planned. 

However, the results remain insignificant. Furthermore, given the current correlation we would need to further increase the sample size substantially, in order to document a significant positive effect of higher stakes; to about 247, as explained in the paper. In other words, the increased total sample size we now rely upon allows us to say that the sample should have shown a correlation of r = 0.16 in order for the results to be statistically significant with a simple linear regression. In other words, we find no significant result, and if there is an effect, it is relatively small – and certainly economically inefficient.

We’d also like to emphasize that our sample size is at the high end, compared to the 9 studies we refer to in our theoretical framework. As in other related theoretical fields, effect sizes of the impact of high (vs. low rewards) seem to turn out smaller than original studies indicate. We should also add that the performance in each condition in the newly collected data is almost identical to our former results, indicating that we should not worry that behavior was different in the Summer of 2020, compared to pre-pandemic times (see S4 Appendix for the relevant table). Our country was functioning quite normally in the relevant period, btw.

All findings, tables and figures have been updated, integrating the new data and points we make in the above section. In accordance with your suggestion, we have maintained the general outline of the paper, but also adjusted some phrasings to ensure that we write, for example; “we do not find a significant difference between the two conditions”, rather than, for example: “our results show that a higher stake does not lead to increased performance”. In essence, we report a null-finding, yet we also document that given our setup, one requires fairly large sample sizes, if one wants to showcase robust significant results.

Again, thank you for your encouraging comments.

Response to Reviewer 2

We would like to thank you for engaging with our paper and challenging our choices and findings. In the following we respond to each of your questions and comments.

Reviewer comment: This paper considers the issue of whether higher stakes lead to better performance. It is a somewhat controversial topic. But to me the main issue is at the end of the range, with very high or very low incentives. This study has payoffs in one case multiplied by five and I am not sure this captures the issue. But it appears the experiments were conducted properly, so it would appear to be rigorous. I just don't see what we learn from this study.

Reply: Thank you for challenging our approach with this comment. Former studies investigating the impact of higher (vs. lower) stakes demonstrate varying results, even across a range of differences in the size of the stakes. We worry that publication biases might shape what (significant) results are available, and, it is certainly not clear what effect sizes one should expect. We decided on a factor of five (medium difference compared to former studies), in order to balance the likelihood of maximizing the difference (cf. the Maxmincon principle, Kerlinger 2006), and improve external validity. In other words, a very high factor is less likely to be useful to implement in an actual organizational context. Therefore, we think it would be relevant to know, if it is difficult to establish an effect when multiplying the stakes with five, which already can be argued to be at the high end of what would be realistic in an organizational context. However, we agree that investigating the challenge at the end of the range (very high or very low) would be highly interesting as well, for theoretical reasons.

Reviewer comment: Abstract, 2nd sentence: Seriously? Then why do teams pay pro athletes so much? Can't really be true as stated.

Reply: We agree that the original phrase was unclear and did not convey our intention meaning. We aim to communicate that observational data indicate that increasing the stakes does not seem to enhance the performance, possibly due to choking effects (as Ariely et al. 2009 speculates). The sentence has been rephrased and we appreciate your attentive eye.

Reviewer comment: Introduction, 2nd paragraph: I really don't agree with this. Higher pay clearly typically leads to higher effort. The issue of interest is really just at the extremes of the pay range.

Reply: In the paragraph in question we do not stipulate that higher pay might lead to higher effort. Rather, we discuss the impact on performance. As mentioned above, empirical studies on performance demonstrate ambiguous effects, some emphasizing choking effects (and thus a negative effect of higher stakes) while others find a positive effect on performance of higher stakes. Incidentally, we could add that our original motivation to engage in this study was Ariely et al.’s (2009) highly cited finding that performance would be lower, if you increase the rewards for completing cognitively mundane tasks. Our results do not corroborate Ariely et al.’s (2009) findings.

Reviewer comment: p. 9, middle: These are examples of extreme pressure, not just pressure. And the golf study is just one study. One wonders how many other studies found no effect. 

Reply: Thank you for making this relevant point; sports are usually about extreme pressure situations. We have rephrased the section to make explicit, that such observational sports studies are looking at more extreme situations than a regular employee typically meets. We also acknowledge that this field could also be influenced by publication bias, hiding no effect findings. 

Reviewer comment: p. 9, lower: It's really quite silly to say that payment per se leads to worse performance.

Reply: We assume the reviewer refers to this part: “[…] some studies find a positive effect on performance of higher stakes compared to lower (Castellan Jr., 1969; Achtziger, et al., 2015; Gneezy & Rustichini, 2000), others find a negative effect (Ariely, et al., 2009; Pokorny, 2008) […]” We consider our choice of words to be in line with the statements made by the original authors who write: “Second, and more importantly, the performance of participants was always lowest in the high payment condition when compared with the low- and mid-payment conditions together.” (Ariely, et al., 2009, p. 458). Furthermore, we also refer to Pokorny (2008, p. 255) “Indeed, subjects who are offered very low incentives perform significantly better than those in the NI and the HI treatments. This provides evidence for a positive effect of very low piece rates on work effort in this context.” Overall, we report the diverging empirical results as well as contrasting theoretical perspectives (micro-economics vs. psychology). We think the fact that empirical results point in different directions and that some imply results that from a micro-economic perspectives appear counter-intuitive, provides further rationale for the relevance of the study.

Reviewer comment: p. 9, penultimate full paragraph: I'm not sure this is an entirely fair review of the literature. 

Reply: We have tried to engage in an extensive search of the literature, including looking at all studies that cite older studies (e.g. Castellan Jr. 1969) or the highly cited studies such as Ariely et al. (2009; 949 citations as of August 2020). We would of course appreciate to be made aware of other, relevant studies.

Reviewer comment: First line of paragraph before Hypothesis 1: But this may well depend on where in the range you are.

Reply: We agree that it would be interesting to focus on (for example) the very high range, but consider this outside the scope of our pre-registered approach. We have adapted the paragraph right before the hypothesis, to make our intent even clearer.

Reviewer comment: Hypotheses 1 and 2: Over the entire range?

Reply: As outlined in the comment above, we can only test our pre-registered hypothesis, i.e. comparing high vs. low stakes involving a difference of a factor of five. We have adapted the paragraph right before the hypothesis, to make our intent even clearer.

Reviewer comment: p. 12, lower: The between- versus within-subject could easily have mattered. Good that you mention this later.

Reply: Thank you for this comment. We agree that this difference is meaningful.

Reviewer comment: p. 13, upper: I don't know that a factor of five will matter in this range for this type of task.

Reply: We also did not know in advance what impact a factor of five would have for this type of task at this level of reward. The difference in reward is 8 DKK (more than 1 Euro) per solved task, leading to some of the highest performing participants to earn more than 60 Euros for their effort. If we had asked students about the difference – after finishing the session – we guess that they would find the difference between (for example) 15 and 75 Euros important. We hope future research can examine different levels of rewards, in order to clarify if the effect is different for different levels of rewards and different factors between the groups. 

Reviewer comment: p. 14, Performance: This is still dubious. I'd also like to see separate analyses.

Reply: We assume that ‘separate analyses’ mean that there should be two regressions; one with solved adding tasks as a dependent variable and one with solved anagram tasks as a dependent variable. We appreciate the skepticism and interest in seeing a more fine-grained analysis. However, we have prioritized to follow the pre-registered analysis plan, rather than risking finding spurious results by running too many statistical tests. We have made the data available (due to privacy reasons, age has been removed from the dataset) and other researchers can therefore make their own statistical tests on the data.

Reviewer comment: p. 14, Intrinsic motivation: I'm not really keen on any of these psych measures, but presumably this was stated in the pre-analysis plan.

Reply: These measures are listed in the pre-registration, and hence we have decided to keep them in the article.

Reviewer comment: Before Figure 4: But it does in fact look like there is a positive difference. I wouldn't necessarily be claiming a big victory here.

Reply: We agree that it is important to transparently document the (in)significance of findings, rather than claiming victories that the data can’t support. We have updated our results following our additional data collection. As you indicate, we did showcase an insignificant difference in the first submission already. The difference is almost identical for the second round of data collection, leading to a smaller yet still insignificant p-value. Given the current correlation between stakes and performance, we would need a sample size of about 247 to identify a significant effect (if relying on a simple linear regression). The effect size is thus relatively small at best, if not random fluctuations.

All findings, tables and figures have been updated. We have adjusted some phrasings to ensure that we write, for example; “we do not find a significant difference between the two conditions”, rather than, for example: “our results show that a higher stake does not lead to increased performance”. In essence, we report a null-finding, yet we also document that given our research design setup, one requires fairly large sample sizes, if one wants to showcase robust significant results. In any case, the data are transparently provided and a future meta-analysis can integrate them.

Reviewer comment: Secondary analysis: 1) I'm not sure how much I trust these self-determination measures. 2) With the Cronbach number so low, I would just ignore this.

Reply: 1) We have updated the Cronbach numbers for our measures, following our additional data collection, and also added a paragraph on the reliability of these scales. 2) The lowest number (focus) increased to 0.55 (from 0.45), while the other remained above 0.70. We agree that the number for focus is very low, yet have chosen to keep them in the article. First, they are described in the pre-registration. Second, Ariely et al. (2009) indicated that the construct could be important, and while our measurement approach might have been ineffective, it can help provide guidance for future researchers’ attempt to measure the construct.

Reviewer comment: End of first paragraph of Discussion: I think the results are unclear here. Yes, they are cautionary.

Reply: We have deliberately phrased our summary in a cautionary manner (as also exemplified in of our former comments), since demonstrating a null effect is by definition difficult. We do find it relevant to provide evidence for that a positive impact of higher stakes would – if it exists – be relatively small (following reflective guidelines in Funder & Ozer 2018, and Gignac & Szodorai 2016), and would thus appear to be ineffective to implement.

Reviewer comment: First line of penultimate paragraph: I agree with this. It's interesting.

Reply: We appreciate your encouraging comment.

References

Kerlinger, F. (2006). Ch4: Research design as variance control, In D. de Vaus (Ed.), Research Design Volume I, (pp. 57-66). London: Sage.

---

## [Decision Letter · Decision Letter 1]

24 Sep 2020

Does the size of rewards influence performance in cognitively demanding tasks?

PONE-D-19-35879R1

Dear Dr. Bergenholtz,

We’re pleased to inform you that your manuscript has been judged scientifically suitable for publication and will be formally accepted for publication once it meets all outstanding technical requirements.

Kind regards,

Christiane Schwieren, Dr.

Academic Editor

PLOS ONE

Additional Editor Comments (optional):

As you can see from the reviewer comments below, Reviewer 2 is not yet entirely happy with how you handle the issue of the relatively small "large" incentives. From my own reading of the paper, I only partially agree with him. For a student, this is a significantly larger incentive, but, of course, there is still the possibility that it might not be enough to reach "choking". Thus, if you think this might make sense, you could discuss a bit more why you consider a null result for a "not that huge" increase in incentives to be sufficient to say that there probably is no negative effect of "large" incentives. Overall, however, I think you have done this with sufficient care and also did avoid overselling on that matter.

Reviewers' comments:

Reviewer's Responses to Questions

**Comments to the Author**

1. If the authors have adequately addressed your comments raised in a previous round of review and you feel that this manuscript is now acceptable for publication, you may indicate that here to bypass the “Comments to the Author” section, enter your conflict of interest statement in the “Confidential to Editor” section, and submit your "Accept" recommendation.

Reviewer #1: All comments have been addressed

Reviewer #2: (No Response)

2. Is the manuscript technically sound, and do the data support the conclusions?

Reviewer #1: Yes

Reviewer #2: Partly

3. Has the statistical analysis been performed appropriately and rigorously? 

Reviewer #1: Yes

Reviewer #2: Yes

4. Have the authors made all data underlying the findings in their manuscript fully available?

Reviewer #1: Yes

Reviewer #2: Yes

5. Is the manuscript presented in an intelligible fashion and written in standard English?

Reviewer #1: Yes

Reviewer #2: Yes

6. Review Comments to the Author

Reviewer #1: (No Response)

Reviewer #2: (No Response)

7. PLOS authors have the option to publish the peer review history of their article (what does this mean?). If published, this will include your full peer review and any attached files.

Reviewer #1: **Yes: **Juergen Huber

Reviewer #2: No

---

## [Editor Report · Acceptance letter]

29 Sep 2020

PONE-D-19-35879R1 

Does the size of rewards influence performance in cognitively demanding tasks? 

Dear Dr. Bergenholtz:

I'm pleased to inform you that your manuscript has been deemed suitable for publication in PLOS ONE. Congratulations! Your manuscript is now with our production department. 

Kind regards, 

on behalf of

Dr. Christiane Schwieren 

Academic Editor

PLOS ONE